# COVID-19 among Health Workers in Germany—An Update

**DOI:** 10.3390/ijerph18179185

**Published:** 2021-08-31

**Authors:** Albert Nienhaus

**Affiliations:** 1Competence Center for Epidemiology and Health Services Research for Healthcare Professionals (CVcare), Institute for Health Services Research in Dermatology and Nursing (IVDP), University Medical Center Hamburg-Eppendorf (UKE), 20246 Hamburg, Germany; albert.nienhaus@bgw-online.de; Tel.: +49-40-20207-3220; 2Department for Occupational Medicine, Hazardous Substances and Health Sciences (AGG), Statutory Accident Insurance and Prevention in the Health and Welfare Services (BGW), 22089 Hamburg, Germany

**Keywords:** COVID-19, occupational health, occupational disease

## Abstract

This is an update of our report on COVID-19 among health and social welfare workers in Germany. Workers’ compensation claims for occupational diseases (OD) are recorded in a standardized database of the Statutory Accident Insurance and Prevention in the Health and Welfare Services (BGW). We analyzed which workers in the health and welfare sector are most often affected by COVID-19. For the different sectors in healthcare and welfare, the number of full-time workers is known (FTW), allowing for calculation of claim rates by sector. The period for data presentation was extended to 3 May 2021 for this update. The cumulative number of COVID-19 claims increased from 4398 by May 2020 to 84,728 by May 2021. The majority of claims concern nursing homes (39.5%) and hospitals (37.6%). Nursing is the profession most often concerned (68.8%). Relative to the number of workers, the claim rate is highest for hospitals (41.3/1000 FTW). Seventy-seven workers died (0.09%) and three hundred and seventy-five (0.4%) were hospitalized. A total of 65,693 (77.5%) claims were assessed, and for 81.4% of these claims, the OD was confirmed. The number of health and welfare workers affected by COVID-19 is high. With most HW vaccinated by now in Germany, within the next few weeks or months, the number of new cases should decrease.

## 1. Introduction

In a pandemic, health workers (HW) are at increased risk for work-related infections. This lesson was already learned from the influenza A (H1N1) pandemic in 2009 and the outbreaks of the severe acute respiratory syndrome (SARS) in 2002 and 2003 as well as the Middle East respiratory syndrome (MERS) outbreaks since 2012 [1,2,3,4,5,6,7,8], and it has been confirmed in this pandemic. By May 2020, 152,888 infections and 1413 deaths of HW due to COVID-19 were reported in literature worldwide [9].

In northern Italy, which was particularly hit by the first COVID-19 wave in Europe, seroprevalence in HW in the high exposure group was 28.5% compared to 12.8% in the low exposure group [10]. Similarly high infection rates are found in HW from Spain. In a hospital-based study, 31.6% of the HW were seropositive for SARS-CoV-2. The odds ratio for infection was highest for doctors (OR = 2.4) [11]. In Germany, the seroprevalence rate of HW after the first wave was as low as 3.3%. Again, HW with contact to COVID-19 patients had an increased risk of infection [12]. According to German health insurance data, sick leave or hospitalization due to COVID-19 was 2.4 times more likely in HW than in all other professions [13].

Not only are HW at increased risk of infection, they also can increase the risk of infection for their families. In a study from Scotland, the hazard ratio for hospitalization due to COVID-19 was 3.3 for HW and 1.8 for their family members [14]. In a French study, the odds ratio for a positive PCR was 3.1 for HW; two relatives died because isolation of the affected HW was not effective. [15]. In a British study, an increased risk was observed not only for HW but also for social workers and teachers (7.4 versus 1.8) [16]. In a study from Washington State, HW had higher infection rates than the other professions [17]. However, it was also shown that infection rates tend to be higher in typical low-wage industries such as farming, fishing, and forestry, or building and grounds cleaning and maintenance.

According to a systematic review and meta-analysis, infected HW are most often female (78.6%). In addition, the authors summarize studies showing that faulty handwashing, inadequate use of masks and personal protection equipment (PPE) or a shortage of PPE increases the risk of infection for HW, while adequate use of PPE reduces the risk of infection [18].

Neither H1N1 nor SARS or MERS led to a significant increase in the number of infections reported as occupational diseases (OD) in Germany [19]. As can be suspected from the brief summary of the actual literature above, during the pandemic, the number of claims concerning infections with SARS-CoV-2 increased in the first five months of 2020, as we reported in our first publication [20]. Here we provide an update on the number of seropositive, COVID-19-related claims filed by HW in Germany by May 2021. In addition, we analyze which workers in the health and welfare sector are most often affected by COVID-19 and suffer severe consequences such as hospitalization and death.

## 2. Materials and Methods

Suspected cases of occupational diseases are recorded in compensation insurance providers “BK-DOK” in a standardized manner. The BK-DOK (or “Berufskrankheiten-Dokumentations-System”) is the database in which all workers’ compensation claims for occupational diseases (Berufskrankheiten-OD) are assessed. In this database, the job title and the particular sector where the worker is engaged are documented. Along with the profession and sector, reporting obligations regarding whether the OD has been confirmed or rejected and whether hospitalization or death have occurred are documented in the system. However, since the database was adjusted in order to assure quick reporting, information on gender and age is not yet available. The number of particular professionals covered by the accident insurance provider is unknown. However, for different sectors (e.g., hospitals or nursing homes) the number of full-time workers (FTW) is known. Two workers working 50% of the usual working time of 39 h per week are considered one FTW. The number of claims per 1000 FTW (claim rate) and the number of hospitalizations and deaths per 100,000 FTW are calculated.

As with all other diseases, a COVID-19 illness which is presumably occupational in nature must be reported to the accident insurance provider or the federal state-level trade office (Landesgewerbeamt) by the physician in charge of the patient concerned. In addition, a patient or a health insurance can file a claim if the disease is suspected to be caused by an exposure at the workplace. There is no timeframe within which the claim needs to be filed, and some delay in reporting infections or other diseases must be taken into account. The system of OD management is mixed in Germany. The government issues a list of occupational diseases with an opening clause stating that a disease which is not yet listed as an OD can be recognized as such if new evidence emerges that it is caused by exposure at the workplace. In total, more than 60 ODs are listed. They are grouped either by kind of exposure (1: chemical, 2: physical, 3: biological) or by organ affected (4: lung, 5: skin). An additional group covers all other ODs; however, in terms of numbers, this sixth group is not important. COVID-19 is caused by a biological agent and therefore belongs to group 3. As an “infection caused by human-to-human transmission”, it has been assigned number OD 3101 [21]. The criteria for recognizing COVID-19 as an OD are explained by Nowak et al. 2021 [22]. In brief, they apply to people working in healthcare, in laboratories testing for SARS-CoV-2, in welfare, or in places with an infection risk similar to that of healthcare. In addition to being compensated for an OD, workers in places not included in the list of ODs can be compensated for a work-related accident caused by COVID-19, e.g., following contact with an infectious colleague at the workplace or after an outbreak in a slaughterhouse.

German accident insurance providers each cover different branches. The BGW is the accident insurance provider for the private health and welfare sector including churches and other NGOs; therefore, the BGW database does not cover workers in state-owned hospitals. As a rule of thumb, it can be assumed that about half of the work-related infections in HW are covered by the BGW. The rest are covered by accident insurance providers in the various German states [23].

The analysis of this data is mostly descriptive. Odds ratios and 95% confidence intervals (CI) are calculated for the hospitalizations and deaths of nurses and physicians, using other HW as a comparison group. As aggregated anonymous data is used, the Helsinki declaration is honored. In this analysis, only cases subject to mandatory reporting are considered. This means the diagnosis of COVID-19 is ascertained by a positive SARS-CoV2-PCR and the infection is symptomatic. In addition, the infection is likely to be work-related.

## 3. Results

A total of 84,728 COVID-19 cases suspected to be work-related were registered by 3 May 2021 by the BGW (Table 1). Most claims concern inpatient and outpatient nursing (39.5%) or clinics (37.6%). As few as 3.2% of the claims concern medical practices. Up to 3 May, 77.5% of all claims were assessed, and in 81.4% of these, the OD was confirmed. For clinics, the assessment rate and the confirmation rate are highest (83.2% and 84.1%) when the small group of “Others” is disregarded. The confirmation rate for inpatient and outpatient care is comparable to the one for clinics (81.8% versus 84.1%), although the assessment rate differed to some extent (76.4% versus 83.2%). As shown in Figure 1, the OD rate per 1000 FTW is highest in clinics, followed by inpatient and outpatient care (28.9 and 20.8). For all other sectors, the OD rate per 1000 FTW is lower, and far below of the average rate of 10.5.

Of the 84,728 claims, 375 (0.4%) workers were hospitalized and 77 died (0.09%). The majority of workers in need of hospital treatment come from clinics (52.3%) and inpatient and outpatient care facilities (28.0%) (Table 2). Correspondingly, the hospitalization rate per 100,000 FTW is highest for clinics (25.4), followed by inpatient and outpatient care (10.5) and medical practices (8.7). Again, the majority of workers who died because of COVID-19 come from clinics or in- and outpatient care (*n* = 22, or 28.6% each). The third most affected sector are facilities for occupational rehabilitation and workplaces for people with disabilities (*n* = 12) (see work with disability in Figure 1). The rate of death per 100,000 FTW was as high as for clinics (2.9), though the hospitalization rate per FTW differed (0.2 versus 25.4).

Most claims concern nurses and nurses’ aides (68.8%), followed by physicians (5.4%) (Table 3). Disregarding “Other professions”, the confirmation rate was lowest in kindergarten teachers (73.6%) and highest in nurses and nurses’ aides (83.7%). Hospitalization most often concerns nurses and nurses’ aides (71.5%), followed by physicians (9.3%). The same is true for deaths resulting from COVID-19 (nurses and nurses’ aides 45.5%, physicians 15.6%).

With physicians and nurses, the odds for a claim concerning hospitalization increased compared to other HW (OR 2.4 and 1.4) (Table 4). For deaths, the odds increased for physicians (OR 1.9) and were lower for nurses (OR 0.4). The effect was statistically significant for nurses but only of borderline significance for physicians (95% CI 1.0–3.8, *p* = 0.06).

## 4. Discussion

COVID-19 has entirely changed the occupational occurrence of infections in Germany. In past years, roughly 800 to 1000 claims of infectious diseases subject to mandatory reporting were submitted to the BGW each year [19]. In the previous publication, which covered the first four months of the pandemic, the BGW had already received 4398 claims due to SARS-CoV-2 and COVID-19 which were subject to mandatory reporting. Eleven deaths and one hundred and fifty-one severe illnesses requiring hospitalization demonstrate the particular vulnerability of healthcare workers [20]. These numbers increased to 84,728 claims, 375 hospitalizations and 77 deaths. This means the number of claims increased more than the number of deaths (19 times versus 7 times for deaths and 2.5 times for hospitalizations). The steep increase of claims is well defined by the three pandemic waves in Germany. The first wave at the beginning of the pandemic was less severe than the second wave starting in October 2020 and the third wave starting in March 2021. The first wave had a maximum of 5000 new cases per day and the second and third waves a maximum of 25,000 and 20,000 per day. However, the different increase in claims compared to hospitalizations and deaths might indicate that more cases of less severe disease were reported. Two factors might explain this development. At the beginning of the pandemic, it was not clear under which terms COVID-19 would be accepted as an OD. This became clearer with a publication explaining the conditions under which COVID-19 is recognized as an OD [23]. Probably more importantly, only during the course of 2020 did it become evident that long-lasting illness can occur even after a light case of the acute infection [24]. Therefore, milder forms of the infection might have been reported to make sure that, in case symptoms occur later, the infection is registered.

The relatively low proportion of severe illness (hospitalization) after COVID-19 (0.4%) can possibly be explained not only by the increased availability of tests but by the fact that our data covers the working age population and younger people, who exhibit severe progressions less frequently than older people [25,26]. However, this could also be explained by the fact that cases are discovered more actively for health workers, in the context of contact tracing or occupational medical check-ups. As a result, cases are discovered which would not have been found otherwise due to a lack of symptoms. After tests became available in Germany at the beginning of the pandemic, HW were tested after they had known contact to COVID-19 patients or when they showed symptoms of COVID-19. It was only in autumn 2020 that employers were obligated to offer regular testing to all workers, regardless of whether they had known contact to COVID-19 patients at the workplace. Though the number of tests performed in the different health and welfare sectors is not available, it is safe to assume that HW in hospitals and nursing homes were tested more frequently than other HW since hospitals and nursing homes were of particular concern. Therefore, the increased number of workers’ compensation claims by HW in hospitals and nursing homes is partly explained by the higher number of tests performed and the resulting higher diction rate of SARS-CoV-2 infection that might have been unnoticed without the regular testing. However, the high number of COVID-19 cases treated in hospitals and the high number of older people infected in nursing homes are a cause of the high infection risk of hospital and nursing home workers. By now, more than 275,000 COVID-19 cases have been treated in hospitals [27]; 132,952 infections were identified in older people over the course of 4937 outbreaks in nursing homes and the resulting contact tracing [28].

Most of the burden of the disease is carried by nurses or nurses’ aides as well as physicians, in terms of both the number of claims and the number of deaths. However, a wide range of other occupations is also concerned. Surprisingly, the rate of accepted claims does not differ much between the different sectors, most likely because workers in all the sectors covered by BGW have close contact to patients or clients. In addition, it should be noted that claims are filed when an OD is suspected and the legal aspects have been taken into account.

Differences between the HW groups in the proportion of claims concerning hospitalization or death might be explained by different screening strategies, or by the varying degree of likelihood that they will file a workers’ compensation claim. However, the higher intensity and frequency of exposure for nurses and physicians might also increase the probability of a severe cause of the disease. In a Swedish study comprising 9500 workers, subjects with high amounts of SARS-CoV-2 virus, as indicated by the cycle threshold (Ct) value in the PCR, had the highest risk for sick leave in the two weeks following testing (OR 11.97 (CI 95% 6.29–22.80)) [29]. However, in a study by Chien et al., it was not confirmed that close contact to an infectious person, e.g., in a household, increases the probability of a high virus load [30].

This analysis is based on workers’ compensation claims relating to COVID-19 in health and welfare workers. In the literature, there are several additional studies available that use secondary data from claim statistics. In the Czech Republic, 150 cases of COVID-19 were recognized as occupational diseases in 2020; of these, 148 were cases in the health and social care sector. By contrast, of 732,202 COVID-19 cases that were registered by December 2020 in the Czech Republic, 34% had work-related contacts [31]. This indicates a large gap between work-related infection risk and recognition as ODs. However, according to the authors, a number of delays exist between the process of recognition of occupational diseases and their entry into the statistics, meaning that a significant proportion of the cases of occupational diseases that are detected in 2020 will be included in the 2021 statistics.

Marinaccio et al. analyzed the workers’ compensation claims collected by the Italian Workers’ Compensation Authority (INAIL) [32]. In the period of March–October 2020, 65,804 compensation claims for COVID-19 were collected by INAIL. The ratio between compensation claim applications and COVID-19 cases in the general population decreased from 20% in March–April to 3–4% in September–October. Most claims concern nurses and paramedics; physicians follow in fourth place. The number of recognized OD is not given. Given the smaller population size, the number of claims in Italy is comparable to the number of claims in Germany. In another analysis from this working group, it is estimated that about 19% of all COVID-19 cases are work-related. The sectors with the highest burden are health and social work; however, occupational compensation claims were also made by meat and poultry processing plant workers, store clerks, postal workers, pharmacists and cleaning workers [33].

Bernacki et al. analyzed about 2000 COVID-19 workers’ compensation claims from 11 states in the U.S. Health and social workers filed 84% of these claims [34]. The proportion of COVID-19-related claims to all claims was as high as 32% in April 2020 and declined to 7% in August 2020. The difficulties that can arise from evaluating workers’ compensation claims related to COVID-19 are detailed by Hayman et al. COVID-19 is a multi-organ disease and assessment of the long-term consequences depends on the organs affected [35]. In addition, as the authors point out, recognition of a claim as an OD depends on the legislation of the different states. Guthrie et al. indicate that the number of workers’ compensation claims due to COVID-19 is low in Australia so far; however, a spike in claims in areas such as geriatric care and the medical and allied professions is expected [36]. This might give rise to legal and practical questions concerning workers’ compensation.

The data presented here do have some limitations. The data of accident insurance provider BGW, generated in view of the reporting obligations of the Occupational Disease Ordinance (Berufskrankheitenverordnung), is likely subject to underreporting. A disease which causes only mild symptoms for a few days might not be reported because the insurance will not grant any compensation for this disease. In addition, only half of the work-related infections in Germany pertain to HW covered by the BGW [23]. Therefore, the numbers of claims of an OD because of COVID-19 should roughly be doubled for an estimate of the real number of affected health and social workers in Germany. However, it should be borne in mind that due to the complicated nature of the disease, COVID-19 is preferably treated in large hospitals with the corresponding expertise [37]. Though a large number of hospital workers (>80%) covered by the BGW work in hospitals with more than 800 HW, in Germany, large hospitals tend to be public and therefore covered by government insurance. Thus, the number of hospital workers affected might even be higher than estimated in this paper. In addition, it should be borne in mind that the particular OD for infection diseases covers health and welfare workers but not most other workers. For other workers, a work-related infection can be compensated as a workplace accident. However, it will be difficult to show that a bus driver, for example, was infected by a passenger, even though in literature an increased infection risk for transportation workers is reported [16]. This is also true for other essential workers who cannot protect themselves by working at home or at a distance; they have an increased infection risk [38]. Short of an outbreak situation, they will have difficulty proving that the source of the infection was at the workplace. Therefore, the burden of infection for workers might be underestimated during the pandemic. As the vaccination campaigns progress not only in HWs, this problem will be mitigated in the near future [39].

## 5. Conclusions

Following the presented data, 53,472 cases of COVID-19 in health and welfare workers have been confirmed as OD by the BGW in Germany. The real number is likely to increase since not all claims have been assessed yet and new claims are still being filed—although a high number of HW and social workers are now vaccinated in Germany. These high numbers demonstrate the importance of preparedness for the next pandemic, and indicate that long-term effects of COVID-19 in workers should be monitored closely.

## Figures and Tables

**Figure 1 ijerph-18-09185-f001:**
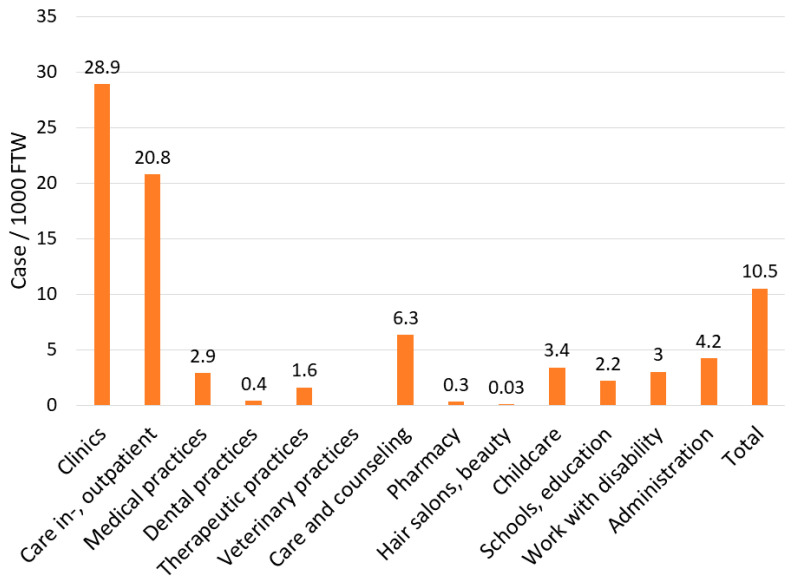
Number of occupational diseases (OD) due to COVID-19 per 1000 full-time workers (FTW) by sector.

**Table 1 ijerph-18-09185-t001:** COVID-19 cases reported to accident insurance provider BGW and confirmed occupational disease (OD) by sector.

Sector	Claims		Assessed		Confirmed OD	
	*n*	% ^a^	*n*	% ^b^	*n*	% ^c^
Clinics	31,818	37.6	26,462	83.2	22,256	84.1
Inpatient and outpatient care	33,458	39.5	25,545	76.4	20,891	81.8
Medical practices	2670	3.2	1817	68.1	1398	76.9
Dental practices	350	0,4	167	47.7	105	62.9
Therapeutic practices	1069	1.3	618	57.8	449	72.7
Veterinary practices	8	0.01	-	-	-	-
Care and counseling	7892	9.3	5826	73.8	4622	79.3
Pharmacy	112	0.1	72	64.3	42	58.3
Hair salons, beauty	49	0.06	13	26.5	7	53.9
Childcare	3760	4.4	2449	65.1	1820	74.3
Schools, education	337	0.4	240	71.2	167	69.6
Occupational rehab and workplaces for people with disabilities	2358	2.8	1879	79.7	1.246	66.3
Administration	827	1.0	582	70.4	455	78.2
Others	25	0.3	15	60.0	13	86.7
Total	84,728	100.0	65,693	77.5	53,472	81.4

^a^ Percentage of all reportable cases. ^b^ Percentage of all cases reported within the sector. ^c^ Percentage of all cases with a decision (assessment) within the sector.

**Table 2 ijerph-18-09185-t002:** Hospitalization or death of COVID-19 cases reported to accident insurance provider BGW by sector.

Sector	Hospitalizations		Hospitalizations/ 100,000 FTW *	Deaths		Deaths/ 100,000 FTW
	*n*	%		*n*	%	
Clinics	196	52.3	25.4	22	28.6	2.9
Inpatient and outpatient care	105	28.0	10.5	22	28.6	2.2
Medical practices	42	11.2	8.7	8	10.4	1.7
Dental practices	1	0.3	0.4	-	-	-
Therapeutic practices	6	1.6	3.1	3	3.9	1.1
Care and counseling	20	5.3	2.7	6	7.8	0.8
Pharmacy	1	0.3	0.7	1	1.3	0.7
Hair salons, beauty	1	0.3	0.5	1	1.3	0.5
Childcare	1	0.3	0.2	1	1.3	0.2
Schools, education	-	-	-	1	1.3	1.3
Occupational rehab and workplaces for people with disabilities	1	0.3	0.2	12	15.6	2.9
Administration	1	0.3	0.9	-	-	-
Others	-	-	-	-	-	-
Total	375	100.0	7.4	77	100.0	1.5

* FTW = full time workers.

**Table 3 ijerph-18-09185-t003:** COVID-19 cases reported to accident insurance provider BGW broken down by profession.

Profession	Claims		Assessed		Confirmed		Hospitalized		Deaths	
	*n*	% ^a^	*n*	% ^b^	*n*	% ^c^	*n*	% ^a^	*n*	% ^a^
Physicians	4547	5.4	3751	82.5	3044	81.2	35	9.3	12	15.6
Nurses, nurses’ aides	58,296	68.8	46,259	79.4	38,705	83.7	268	71.5	35	45.5
Medical assistant	3548	4.2	2551	71.9	1987	77.9	36	9.6	2	2.6
Physiotherapist	1390	1.6	1003	72.2	832	83.0	8	2.1	1	1.3
Social worker	3256	3.9	2413	74.1	1790	74.2	4	1.1	2	2.6
Kindergarten teacher	4280	5.1	2809	65.6	2068	73.6	5	1.3	1	1.3
Housekeeping, Cleaning	2977	3.5	2198	73.8	1671	76.0	5	1.3	5	6.5
Other	6434	7.6	4709	73.2	3375	71.7	14	3.7	19	24.7
Total	84,728	100.0	65,693	77.5	53,472	81.4	375	100.0	77	100.0

^a^ Percentage of all reportable cases, all hospitalized or all deaths. ^b^ Percentage of all cases reported within the sector. ^c^ Percentage of all cases with a decision (assessment) within the sector.

**Table 4 ijerph-18-09185-t004:** Workers’ compensation claims concerning hospitalization and death with odds ratios (OR) and confidence interval (CI) broken down by nurses, physicians and other health workers (HW).

Profession	Hospitalized			Deaths		
	Yes	No	OR (95%CI)	Yes	No	OR (95%CI)
Other HW	72	21,813	-	30	21,783	-
Physicians	35	4507	2.4 (1.6–3.5)	12	4535	1.9 (1.0–3.8)
Nurses	268	58,028	1.4 (1.1–1.8)	35	58,261	0.4 (0.3–0.7)

## Data Availability

The data are available from the author upon request.

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
