# Peer review of "COVID-19 among Health Workers in Germany—An Update"

_ijerph, 2021, doi:10.3390/ijerph18179185_

Round 1

Reviewer 1 Report

This study provides a good understanding of the changes in COVID-19 among health workers in germany after COVID-19. It is necessary to consider the following points.

First, the analysis is too simple description. In Germany, from paper, we well know the situation about COVID-19 of health workers and the result of claim. That's all. It is necessary to construct a causal relationship model and to verify it.

Second, what is the theoretical basis of this study? Is it an institutional approach or an approach from an epidemic perspective? It is necessary to describe the results of this study in terms of previous or theoretical studies and frames.

Third, it is recommended to approach this analysis comparatively. Do you have any data to compare the findings from this study with those from other countries?

Author Response

I am gratefull for the thoughtful comments of the reviewer and made revisions and admentments accordingly as can be seen below.

Reviewer 1

This study provides a good understanding of the changes in COVID-19 among health workers in germany after COVID-19. It is necessary to consider the following points.

1) First, the analysis is too simple description. In Germany, from paper, we well know the situation about COVID-19 of health workers and the result of claim. That's all. It is necessary to construct a causal relationship model and to verify it.

Authors answer: Thank you for this suggestion. Unfortunately, we do have aggregated data. But more important, some important information for causal inferences are missing like gender, age and number of workers in the different jobs. The causal inference we can draw is that workers in hospitals and nursing homes are more often concerned then other health and social workers. This we now point out more clearly. In addition, relative to the number of claims, the risk of hospitalisation and death is higher for physicians than for other HW. I included a new table (table 4) and calculated Odds Ratios. Now potential reasons for this observation (underreporting of claims for physicians or more intensive exposure) are discussed.

 2) Second, what is the theoretical basis of this study? Is it an institutional approach or an approach from an epidemic perspective? It is necessary to describe the results of this study in terms of previous or theoretical studies and frames.

Authors answer: This is a secondary data analysis using available data of the BGW. Therefor I think it is an institutional approach. We assume that HW in hospitals and nursing homes are carrying the highest burden of disease as this can be deduced from the literature overview and which is supported by our data. Now we state this more clearly in the introduction. I think compensation of HW infected at the workplace is a question of justice and social fairness. See also answer to your comment number 3. 

3) Third, it is recommended to approach this analysis comparatively. Do you have any data to compare the findings from this study with those from other countries?

Authors answer: Thank you for this comment. I searched literature accordingly for secondary data from workers compensation claims from accident insurances. Using the key words COVID-19, occupational disease or workers’ compensation claims I found articles from the US, Spain, Italy and Czech Republic and Australia. All paper confirm that HW are the most affected workers. However, the number of claims accepted as OD differs and is very low in Czech Republic. The assessment of claims is difficult. This is now discussed in more detail in the discussion.

Reviewer 2 Report

The manuscript covers an important field of health service research in times of the COOID-19 pandemic.  Actually, during the debate of mandatory vaccination for health care workers (HW) the information about the number of seropositive HW, infections and death of HW due to COVID-19 is important. Unfortunately, the results section is somewhat difficult for the reader to follow and a major limitation of the study is not stated out clearly enough. Therefore, I kindly ask the author to work on the following detailed  comments before publishing.

Line 40/41           the authors

"…. The authors…."  Only one author is mentioned on the title page.

Line 46

Maybe the author should ad, that the number of seropositive HW is mentioned.  

Line  50                BK-DOK

Please explain “BK-DOK“ to the reader.

Line 62/63           Occupational disease 3101

The German system of occupational diseases may not be familiar to the international readership of the journal. I recommend explaining this in one or two sentences.

Line 66 – 70        Database            

The author states, that the database of the BGW does not cover workers in state-owned hospitals. IN my opinion, this is a relevant selection bias of this study.

Significantly more than half of the COVID-10 patients were treated in hospitals with more than 400 beds or even maximum care hospitals. However, these (especially the maximum care hospitals) are state-owned. In particular, severe cases of COVID-19 in Germany were transferred to maximum-care hospitals at an early stage. Therefore, exposure to COVID 19 will have been different for staff in government hospitals than in non-government, respectively smaller, hospitals. This should be clarified by the author. Even in the discussion section, it should be discussed in more than one sentence. 

https://www.bundesgesundheitsministerium.de/fileadmin/Dateien/3_Downloads/C/Coronavirus/Analyse_Leistungen_Ausgleichszahlungen_2020_Corona-Krise.pdf

Line 75 – 114

In the text paragraph of the result section, the author provides the same information, what the tables show to the reader. Please try to avoid redundancy. Maybe some information can be provided by figures?

L133-135 working age and severe illness

If this data is available, please provide them and correlate (correlation or regression analysis, whatever is appropriate) them.

L 135-137 contact tracing – medical check ups and testing bias.

Please provide references for  this assumptions.

L 138-139 claims of nurses in comparison to physicians are more than tenfold higher and death threefold higher. Therefore, a distinction should be made.

Line 158 [Agnessa]

I guess the Author wanted to cite his co-wroker Dr. Agnessa Kozak?

Author Response

I am thankful for the comments and suggestions of reviewer 2. I revised the manuscript accordingly as can be seen below. (As Reviewer I and three comments, I started the numbering with four.)

Reviewer general comment

The manuscript covers an important field of health service research in times of the COOID-19 pandemic.  Actually, during the debate of mandatory vaccination for health care workers (HW) the information about the number of seropositive HW, infections and death of HW due to COVID-19 is important. Unfortunately, the results section is somewhat difficult for the reader to follow and a major limitation of the study is not stated out clearly enough. Therefore, I kindly ask the author to work on the following detailed  comments before publishing. 

4) Line 40/41           the authors

„…. The authors….“  Only one author is mentioned on the title page.

Author’s answer: ‘The authors’ refers to the authors of the review [18]:

  • Gholamia, M.; Fawada, I.; Shadana, S.; Rowaieea, R,; Ghanema, H.A.; Khamisb, A.H.; Hoa, S.B. COVID-19 and healthcare workers: A systematic review and meta-analysis. International Journal of Infectious Diseases 2021;104:335-346

5) Line 46 Maybe the author should ad, that the number of seropositive HW is mentioned.  

Authors answer: Thank you for this comment. We made changes accordingly: Here we report an update on the number of seropositive, COVID-19-related claims filed in Germany until May 2021 from HW.

6) Line  50                BK-DOK Please explain “BK-DOK“ to the reader.

Authors answer: Thank you, now I explain: The BK-DOK is the “Berufskrankheiten-Dokumentations-System”. This is the standardized data base, in which all workers compensation claims for Occupational Diseases (Berufskrankheiten) are assessed.

7) Line 62/63           Occupational disease 3101

The German system of occupational diseases may not be familiar to the international readership of the journal. I recommend explaining this in one or two sentences.

Authors answer: Thank you for this suggestion. Now I briefly explain the German system of occupational diseases. In Germany the system of occupational diseases is a mixed system. There is a list of occupational diseases, which is issued by the government and in addition there is an opening clause stating that a disease so far not listed as OD can be recognized as an OD if new evidence emerges that the particular disease is caused by exposure at the workplace. In total more than 40 ODs are listed. They are grouped either by kind of exposure (1 chemical, 2 physical, 3 biological) or by organ effected (4 lung, 5 skin). An additional group covers all other OD. But in terms of numbers this sixth group is not important. Covid-19 is caused by a biologic agent and therefore belongs to group 3. In particular, it is called: ‘Infections caused by human to human transmission’ and has the number OD 3101 The criteria for recognizing COVID-19 as an OD are explained by Nowak et al. 2021. In breve, working in healthcare, social welfare, laboratories testing for SARS-CoV-2, or working in places with infection risk similar to healthcare is a prerequisite for recognizing COVID-19 as OD. In addition for being compensated for an OD, workers in other workplaces not included in the list of ODs, can be compensated for a work related accident caused by COVID-19, e.g. after contact to an infectious college at the workplace or after an outbreak in a slaughterhouse.    

8) Line 66 – 70        Database            

The author states, that the database of the BGW does not cover workers in state-owned hospitals. IN my opinion, this is a relevant selection bias of this study.

Significantly, more than half of the COVID-10 patients were treated in hospitals with more than 400 beds or even maximum care hospitals. However, these (especially the maximum care hospitals) are state-owned. In particular, severe cases of COVID-19 in Germany were transferred to maximum-care hospitals at an early stage. Therefore, exposure to COVID 19 will have been different for staff in government hospitals than in non-government, respectively smaller, hospitals. This should be clarified by the author. Even in the discussion section, it should be discussed in more than one sentence. 

https://www.bundesgesundheitsministerium.de/fileadmin/Dateien/3_Downloads/C/Coronavirus/Analyse_Leistungen_Ausgleichszahlungen_2020_Corona-Krise.pdf

Authors answer: This is a very good point. However, it is no longer true that maximum care hospitals in Germany are predominately state owned and therefor insured by the state owned accidence insurances. We do not have the number of beds available for the hospitals covered by the BGW but the number of HW per hospital. More than 80 % of the hospital workers work in hospitals with more than 800 workers. From this, it becomes evident that maximum care hospitals are also covered by the BGW. Now we discuss this in more detail. In addition, yes you are right it would be good to have an overview over of workers’ compensation claims concerning COVID-19 in Germany. My working group is trying to get our hands on these data. However, this will take some time.

9) Line 75 – 114

In the text paragraph of the result section, the author provides the same information, what the tables show to the reader. Please try to avoid redundancy. Maybe some information can be provided by figures?

Authors answer: In the results section we do not provide all the number of the tables but the main numbers. We carefully checked which numbers can be omitted in the text. Now we provide a figure on the number of case per 1,000 full-time workers.

10) L133-135 working age and severe illness

If this data is available, please provide them and correlate (correlation or regression analysis, whatever is appropriate) them.

Authors answer: This is an interesting point. With the data set used we do not have information regarding age.

11) L 135-137 contact tracing – medical check ups and testing bias.

Please provide references for  this assumptions.

Authors answer: Thank you for this point. Now we describe the test strategy for HW and other workers in Germany and we researched literature

12) L 138-139 claims of nurses in comparison to physicians are more than tenfold higher and death threefold higher. Therefore, a distinction should be made.

Authors answer: Thank your for the comment. This is an important point. As we do not know how many nurses or physicians are covered by the BGW we cannot deduce whether nurses have a higher or lower infection risk than physicians. However, compared to the number of claims, the number of hospitalization and death differ between nurses and physicians. This might either be because Covid-19 with moderate symptoms is underreported in physicians or severe cause of COVID-19 is more likely in physicians (see previous paper).  In order to illustrate the difference we calculated Odds Ratios for claims with hospitalization or death comparing physicians or nurses with all other HW. For physician the odds for hospitalization or death are increased. See new table 4 

13) Line 158 [Agnessa]

I guess the Author wanted to cite his co-wroker Dr. Agnessa Kozak?

Authors answer: Thank you for pointing this out. Your guess is correct. The sentence was revised accordingly

Round 2

Reviewer 1 Report

I can not find any reasons that change the first review decision. 

Author Response

I am sorry that I was not able to convince you. I think the manuscript improved and I think it is important to learn how many HW are affected by COVID-19 and that they have a right to be compensated for their suffering.